# Effect of Proximity to Failure in Resistance Training on Circulating Levels of Neuroprotective Biomarkers

**DOI:** 10.3390/biology14121756

**Published:** 2025-12-07

**Authors:** Brian Benitez, Matthew C. Juber, Christian T. Macarilla, Zac P. Robinson, Joshua C. Pelland, Jacob F. Remmert, Seth R. Hinson, Nishant P. Visavadiya, Michael C. Zourdos

**Affiliations:** 1Muscle Physiology Laboratory, Department of Exercise Science and Health Promotion, Florida Atlantic University, Boca Raton, FL 33431, USA; bbenitez@stetson.edu (B.B.);; 2H.A.T.T.E.R (Human Advancement Through Translational Exercise Research) Lab, Department of Health Sciences, Stetson University, Deland, FL 33431, USA; 3Department of Medicine, Division of Cardiology, University of Colorado School of Medicine, Aurora, CO 80045, USA; 4Consortum of Fibrosis Research & Translation, University of Colorado School of Medicine, Aurora, CO 80045, USA; 5Department of Physiology and Cell Biology, University of South Alabama College of Medicine, Mobile, AL 36608, USA

**Keywords:** neuroprotection, brain-derived neurotrophic factor, humans

## Abstract

Resistance training benefits both muscles and the brain, but researchers don’t fully understand how hard people need to train to achieve brain health benefits. This study investigated whether training to complete exhaustion is necessary to increase brain-protective molecules (brain-derived neurotrophic factor, cathepsin B, insulin-like growth factor-1, and interleukin-6) in the blood. Thirty-eight resistance-trained men were divided into four groups that trained with different levels of effort: some stopped well before exhaustion (4–6 repetitions in reserve), others trained closer to exhaustion (1–3 repetitions in reserve), and one group trained to complete exhaustion on every set. Repetitions in reserve is a way of measuring how hard someone is working during exercise, specifically, how many more repetitions someone perceives they can perform. For example, if someone lifts a weight 8 times and feels they could have performed 2 more repetitions, they have 2 repetitions in reserve. All participants performed resistance training three times weekly for eight weeks. Blood samples were collected before and after workouts to measure four molecules linked to brain health and protection against age-related cognitive decline. The key finding was that all training groups, including those who stopped well before exhaustion, showed similar increases in important brain-protective molecules after exercise. Additionally, it was discovered that resistance training can influence a specific molecule called cathepsin B, which had never been shown before. These results suggest people can gain brain-health benefits from resistance training without pushing themselves to exhaustion, making this type of exercise safer, more enjoyable, and requiring less recovery time.

## 1. Introduction

Age-related cognitive decline poses an inevitable constraint on human healthspan, with measurable decrements in processing speed, executive function, and memory beginning in the third decade of life and accelerating thereafter [1,2]. These functional changes reflect underlying neurobiological alterations, including reduced hippocampal volume, decreased synaptic density, and impaired neuroplasticity [3,4,5]. While the pace and severity of decline vary between individuals, the trajectory is universal: cognitive capacity diminishes with age even in the absence of pathology [1,2].

Despite impaired cognition with aging [1,2], meta-analyses have demonstrated both aerobic (AT) [6] and resistance training (RT) [7] to attenuate the age-related decline in global cognition, memory, executive function, and processing speed. These cognitive benefits are supported by evidence of neurobiological adaptations that preserve neural integrity across aging, including enhanced neurotrophin expression [8,9], increased synaptic plasticity [10], and preservation of hippocampal volume [8,9]. While both AT and RT can promote positive cognitive adaptations, the expression and modulation of key neuroprotective biomarkers have been much more thoroughly researched with AT [11].

Brain-derived neurotrophic factor (BDNF), a neurotrophin involved in synaptic plasticity and neuronal survival, exemplifies the mechanistic disparity between AT- and RT-induced responses. BDNF has been proposed as a molecular mediator linking physical exercise to central nervous system adaptations [8,9,12], and because circulating BDNF can cross the blood–brain barrier [8,9,12], changes in peripheral concentrations are purported to reflect central availability. Indeed, increased circulating BDNF concentrations have been associated with increased hippocampal volume and cognitive performance [8,9], while reduced concentrations are observed in neurodegenerative disease and may predict cognitive decline [13,14]. Acute AT has consistently led to transient increases in circulating BDNF, while chronic AT has consistently produced increases in resting concentrations [11,15]. In contrast, some studies have shown acute RT to lead to transient increases in BDNF [16,17,18] while others have observed no significant change [19,20,21]. Moreover, although long-term RT has improved cognition [7], meta-analyses have failed to observe that chronic RT leads to changes in resting BDNF [11]. However, a cross-sectional study from De la Rosa et al. [22] found that young men who regularly engaged in sports training and middle-aged rugby players had lower resting BDNF compared with sedentary controls, along with better immediate memory performance in the rugby cohort. Although the trained groups were not screened specifically for RT, it is reasonable to assume that their routines included it, particularly among the rugby players; therefore, a contribution from RT cannot be excluded.

The inconsistent findings for RT to induce a BDNF response may be related to training prescription variables such as sufficient proximity to failure of sets defined by repetitions in reserve (RIR), exercise selection, and engagement of large musculature. For instance, studies reporting acute BDNF increases have predominantly utilized multi-joint free-weight protocols (e.g., back squat and bench press) performed to or near momentary muscular failure [16,17,18]. In contrast, protocols prescribing machine-based fixed-repetition schemes (e.g., 3 sets of 10 at 80% of 1RM) [20] or isokinetic single-joint exercises [19,21] have reported no change in circulating BDNF.

It is possible that the increases in BDNF related to protocols combining higher volume and greater intensity are due to interactions with other biomarkers. Specifically, cathepsin B (CatB), a cysteine protease shown to promote neurogenesis through BDNF upregulation in animal models [23], has received limited investigation in RT contexts. Insulin-like growth factor-1 (IGF-1), which is implicated in neural plasticity and cognitive function [24,25], shows variable responses to RT but, similar to BDNF, is increased with more challenging training [24]. Additionally, IGF-1 has been shown to mediate genes expressing BDNF [25]. Moreover, neuroendocrine and immune crosstalk may be important for exercise-induced BDNF elevations as the myokine interleukin-6 (IL-6) has stimulated BDNF production in human monocytes [26], and IL-6 is consistently elevated in response to acute RT [27]. Consequently, the isolated manipulation of proximity to failure may clarify whether these biomarkers respond in concert with or independently of BDNF, potentially revealing multiple pathways through which RT influences neuroprotection.

Therefore, the primary aim of this study was to compare acute and chronic responses of serum BDNF, CatB, IGF-1, and IL-6 across four volume-equated RT protocols differing in proximity to failure (4–6 RIR, 1–3 RIR, 0–3 RIR, and 0 RIR) over an eight-week intervention in resistance-trained men. We hypothesized that there would be a dose–response relationship between proximity to failure and acute biomarker responses. An additional exploratory aim was to examine potential correlations among biomarker responses to identify shared regulatory pathways or compensatory relationships within the neuroprotective response to RT.

## 2. Methods

### 2.1. Experimental Design

This investigation employed a between-participant design with stratified randomization, comparing acute (before and immediately after a single exercise session) and chronic (changes from baseline to post-study) changes in serum BDNF, CatB, IGF-1, and IL-6 across four RT protocols differing in proximity to failure. The eight-week intervention was conducted in resistance-trained men with participants counterbalanced by baseline relative strength and assigned to groups defined by repetitions-in-reserve: 4–6 RIR, 1–3 RIR, 0–3 RIR (targeting 1–3 RIR with the final set on each exercise of each session to failure), and 0 RIR (all sets to failure). Specifically, RIR is a subjective assessment of proximity to failure. For example, if a participant reports a RIR of 3, that means the participant perceives they could have successfully completed three more repetitions before reaching momentary muscular failure.

The training protocol consisted of the back squat and bench press performed on three non-consecutive days per week (e.g., Monday, Wednesday, and Friday). Baseline assessments occurred 48–72 h before week 1 and included 1RM, venous blood sampling, and anthropometric measurements. Post-intervention assessments employed identical procedures 48–72 h following week 8. Blood was collected immediately before and after the first weekly training sessions of weeks 1 and 7. The intervention comprised a group-specific introductory microcycle (week 1), primary training phase (weeks 2–7), and taper (week 8).

Peri-exercise nutrition was standardized with branched-chain amino acid supplementation (3.5 g leucine, 1.75 g isoleucine, 1.75 g valine, 2.5 g glutamine; Core Nutritionals, Arlington, VA, USA) administered 30 min pre-exercise and whey protein (30 g providing 3.5 g leucine; Core Pro, Core Nutritionals) consumed immediately post-exercise. Supplements were provided as powder mixed with 10 oz water. On blood collection days, supplementation occurred after post-session venipuncture. Participants discontinued all other supplementation throughout the investigation.

The protocol received approval from the Institutional Review Board for Human Subjects (IRB #: 1422879-3) and adhered to the Declaration of Helsinki ethical standards. This manuscript represents a subset of a larger investigation with additional outcomes reported elsewhere [28,29].

### 2.2. Participants and Sample Size Justification

Sample size determination was constrained by feasibility [30], specifically the availability of resources and the timeframe required for project completion within graduate training requirements. As a result, an a priori power analysis was not performed. To promote transparency and to facilitate future meta-analytic aggregation of the present data, all data from this study have been made publicly available (https://osf.io/hmjrg/?view_only=6d06b111bc4e446eab41fad7c862c4ed (accessed on 5 October 2025).

Thirty-eight resistance-trained men (18–40 years) were recruited with inclusion criteria comprising: minimum two years structured RT (verified by questionnaire), back squat 1 RM ≥ 1.25× body mass, and bench press 1 RM ≥ 1.0× body mass. Exclusion criteria included medical contraindications identified through health history questionnaires (e.g., heart disease, hypertension, diabetes).

Six participants discontinued due to training-related discomfort or injury (three 0–3 RIR, three 0 RIR). The 0 RIR group was terminated early for safety considerations; retained data require cautious interpretation given the limited sample size. The 0–3 RIR group underwent mid-intervention protocol modification (lower-body prescription changed to 4–6 RIR for >50% of sessions) and was excluded from analysis. Two additional 4–6 RIR participants were excluded (one for required training modifications, one for declining venipuncture due to discomfort with blood draws). A single 0–3 RIR participant and a single 1–3 RIR participant also declined venipuncture, and therefore did not contribute blood biomarker data. Group-specific baseline characteristics are presented in Table 1.

### 2.3. Training Protocol

Training frequency consisted of three non-consecutive weekly sessions with matched set and repetition schemes across groups, differing only in proximity to failure via perceived RIR. Week 1 employed a group-specific introductory microcycle with reduced volume and conservative RIR targets. Initial load selection by investigators utilized participant-reported RIR, barbell velocity, and observed set difficulty, with decision rationale communicated with participants to facilitate prescription understanding. Subsequent load selection became participant-directed within assigned RIR parameters. Additionally, although RIR is subjective, recent data have indicated that individuals with multiple years of training experience, such as in the current study, can predict RIR with acceptable accuracy [31].

The primary training phase (weeks 2–7) employed weekly undulating periodization: weeks 2–3 prescribed 10, 8, 6 repetitions across the three weekly sessions, weeks 4–5 prescribed 9, 7, 5 repetitions, and weeks 6–7 prescribed 8, 6, 4 repetitions. The final training week (week 8) was a taper microcycle where participants performed sets with the average load lifted throughout the preceding weeks for that session (e.g., 7 weeks of session 1 loads averaged). Session 1 consisted of 2 sets of 4 repetitions, and session 2 consisted of 2 sets of 2 repetitions. Participants reported RIR following each set and rested 3–5 min between sets.

To support recruitment and improve ecological validity, all participants also performed assistance exercises with a common RIR prescription. Specifically, all accessory exercises (barbell overhead press, barbell row, barbell lying triceps extension, barbell curl, dumbbell lateral raise) were performed to 2 RIR per set across all groups. A controlled eccentric tempo with participant-preferred concentric velocity was employed throughout. A detailed breakdown of the training protocol, including session-by-session prescriptions and load progression schemes, is publicly accessible via the Open Science Framework (https://osf.io/hmjrg/?view_only=6d06b111bc4e446eab41fad7c862c4ed (accessed on 5 October 2025).

Throughout the training protocol, failure was operationalized as the inability to complete full-range repetition despite maximal effort or volitional set termination. Thus, momentary failure sets, defined as actually failing on a repetition, were coded −1 RIR [32]; volitional termination was coded 0 RIR.

### 2.4. Training Load Instructions and Adjustments

Pre-session instructions were standardized via the following script while displaying the RIR scale:

“Please view this scale to remind you of how RIR is scored. Today, working sets should fall within the RIR range of the insert RIR range assigned for the week. Use your knowledge of your prior performances and how the warm-up sets felt to select a load you believe will fall within the assigned RIR range. The goal is to maintain your loads in a subsequent fashion; therefore, if the load you select falls above or below the target RIR range, an increase or decrease in load will occur on the next set. If you fall within the target RIR range, you have the freedom to increase or decrease load as you see fit, so long as you believe this modified load will still fall within the target RIR range. Avoid being overly conservative or aggressive in your load selection and expect your RIR to rise with each set as fatigue accumulates.”

Intra-session load adjustments for the 4–6 RIR and 1–3 RIR groups were performed in accordance with established procedures [33]. Specifically, if the perceived RIR fell outside the assigned range, the load for the next set was increased or decreased by 2% for every 0.5 RIR from the midpoint of the target range, which was 5 for the 4–6 prescription and 2 for the 1–3 prescription. Completing a set with more than the prescribed RIR prompted an increase, while fewer than prescribed prompted a decrease.

For the 0–3 RIR group, all sets except the final set used the same adjustment rule as the 1–3 RIR group. Before the final set of sessions in weeks 2 through 7, participants in the 0–3 RIR group were instructed to select a load expected to produce failure immediately after the prescribed repetition target, for example, failing on repetition 11 when the target was 10. If the second-to-final set was within 1–3 RIR, the participant selected the final-set load based on that guidance. If the prior set was outside 1–3 RIR, the final-set load was maintained or increased by 2% for every 0.5 RIR from the target of 0 RIR on the upcoming set.

For the 0 RIR group, adjustments were based on repetitions achieved relative to the prescription and were informed by pilot testing in five individuals conducted before data collection. To account for fatigue induced by training to failure, the load from set 1 to set 2 was reduced by 2 percent regardless of whether the target repetition number was met. This fatigue reduction was not applied after set 2. In addition, a 1 percent load change per repetition difference from the target was applied set to set, with increases when repetitions exceeded the target and decreases when repetitions fell short. If target repetitions were achieved on the first set, the load for the second set was reduced by 2 percent. If target repetitions were achieved on sets 2 through 4, the load was held constant. All adjustment rules were applied identically to the back squat and bench press.

### 2.5. Anthropometric Assessments

Body mass was measured via a calibrated digital scale (±0.1 kg; Mettler Toledo, Columbus, OH, USA). Standing height was measured with a wall-mounted stadiometer (±0.1 cm; SECA, Hamburg, Germany). Body composition estimation utilized a three-site skinfold assessment (chest, abdomen, anterior thigh) with Jackson and Pollock equation [34]. Duplicate measurements were obtained per site; discrepancies > 2 mm prompted a third measurement, with the closest values averaged. A single investigator performed all assessments.

### 2.6. Back Squat and Bench Press Technique

The back squat and the bench press were performed in accordance with International Powerlifting Federation standards. For the back squat, participants assumed an erect starting position with hips and knees fully extended and the barbell positioned across the upper back. On the investigator’s “squat” command, participants descended under control until the hip crease passed below the superior border of the patella, then returned to full extension. The “rack” command signaled re-racking. For the bench press, participants assumed a supine position on a flat bench and maintained five points of contact throughout the lift (head, shoulders, and buttocks on the bench, both feet flat on the floor). After achieving full elbow extension in the start position, the “start” command was issued. Participants lowered the bar to make contact with the chest, then pressed to full elbow extension. No pause on the chest was required. The “rack” command signaled re-racking.

### 2.7. One-Repetition Maximum (1 RM) Testing

All 1 RM testing was performed in accordance with previously validated procedures [32]. Participants completed a 5 min dynamic warm-up, then a lift-specific warm-up: unloaded barbell for as many repetitions as desired, 20 percent of estimated 1 RM for 5 repetitions, 50 percent for 3, 70 percent for 2, and 80 percent for 1. After 3 to 5 min of rest, a final warm-up at 85 to 90 percent of the estimated 1 RM was performed. Following 5 to 7 min of rest, the first 1 RM attempt was selected by the investigators. Load was increased between attempts until a 1 RM was achieved, with 5 to 7 min of rest between attempts. After squat 1 RM testing, participants rested for 10 min and then completed the identical protocol for the bench press. A 1 RM was accepted as valid if any one of the following conditions was met: (a) the participant reported an RIR of 0 and investigators judged a heavier successful attempt unlikely; (b) the participant reported an RIR of 0.5 and failed the next attempt with a load increase of 2.5 kg or less; or (c) the participant reported an RIR of 1 and failed the next attempt with a load increase of 5 kg or less. Eleiko barbells and calibrated plates (Chicago, IL, USA) accurate to 0.25 kg were used for all testing.

### 2.8. Blood Sampling and Analysis

Venous blood samples were obtained from the antecubital vein by a member of the research team formally trained in venipuncture, using serum separator tubes and standard phlebotomy procedures. Participants were seated during all venipuncture procedures. Following collection, samples were allowed to clot at room temperature for 20–30 min and subsequently centrifuged at 2000× *g* for 10 min. The resulting serum was aliquoted into cryovials and stored at −80 °C until batch analysis. Concentrations of BDNF, CatB, IGF-1, and IL-6 were analyzed in duplicate using commercially available ELISA kits (Abcam, Cambridge, UK) according to the manufacturer’s instructions, with absorbance measured on an Epoch microplate spectrophotometer (BioTek Instruments, Winooski, VT, USA). The specific kits employed were: Cathepsin B (ab119584, Abcam, Cambdridge, UK), BDNF (ab212166, Abcam, Cambdridge, UK), IGF-1 (ab100545, Abcam, Cambdridge, UK), and IL-6 High Sensitivity (ab46042, Abcam, Cambdridge, UK). Blood samples were collected immediately prior to pre- and post-testing sessions (weeks 1 and 8) and immediately before and after the first training session of weeks 2 and 7.

### 2.9. Statistical Analysis

All statistical analyses were conducted within a Bayesian framework using R version 4.3.0 (R Core Team, Vienna, Austria). The brms package (v2.23.0) served as the primary interface for Bayesian multilevel modeling via Stan’s Hamiltonian Monte Carlo sampling algorithms. Post-processing and visualization were performed through the tidybayes (v3.0.7) and marginaleffects (v0.31.0) packages. Complete analytical workflows, including raw data, model specifications, posterior distributions, and visualization scripts, are publicly accessible via the Open Science Framework (https://osf.io/hmjrg/?view_only=6d06b111bc4e446eab41fad7c862c4ed (accessed on 5 October 2025).

Bayesian multilevel models were constructed with weakly informative regularizing priors to incorporate minimal prior knowledge while maintaining computational stability. Specifically, fixed effect parameters (intercepts and slopes) were assigned normal prior distributions centered at zero with unit standard deviation (Normal [0, 1]). Random effect variances followed exponential distributions with scale parameter λ = 1. Correlations among random effects were governed by the Lewandowski-Kurowicka-Joe (LKJ) distribution with shape parameter η = 4.

Model estimation employed four independent Markov Chain Monte Carlo chains, each executing 1000 warm-up iterations for adaptation followed by 4000 sampling iterations, yielding 16,000 total posterior draws for inference. Convergence diagnostics included visual inspection of trace plots for chain mixing and stationarity, alongside examination of posterior predictive distributions to assess model adequacy and identify potential misspecification.

To quantify the influence of proximity to failure on acute biomarker expression (BDNF, CatB, IGF-1, IL-6), we specified separate Gaussian linear mixed-effects models incorporating the temporal dynamics of the training intervention. Fixed effects comprised main effects and interactions among Week (initial vs. final training week), Session (pre- vs. post-exercise), and Average RIR (operationalized as a continuous covariate).

Participant-specific random intercepts accommodated the repeated-measures structure, while maximal random effects structures were implemented where computationally feasible to account for individual variation in temporal trajectories. Both outcome variables and the RIR predictor underwent standardization to facilitate intuitive prior elicitation and enhance interpretability of effect magnitudes.

Evaluation of training-induced chronic adaptations employed parallel linear mixed-effects models examining baseline-to-post-intervention changes. Fixed effects included Time (pre- and post-intervention) and its interaction with average RIR (continuous), enabling assessment of dose–response relationships between training effort and biomarker adaptations. Random intercepts were introduced to accommodate participant-specific baseline differences, with maximal random slopes incorporated where supported by the data structure. Standardization procedures mirrored those employed for acute analyses.

Parameter inference utilized the complete posterior distribution to generate probability density functions for each effect of interest. Modal estimates (posterior modes) served as point estimates, accompanied by 95% highest density intervals (HDI) to quantify uncertainty. Practical significance was evaluated through comparison with regions of practical equivalence (ROPE) defined by the typical error of measurement for each biomarker, calculated from duplicate sample variability at each timepoint.

Two complementary probability metrics guided interpretation: (1) the probability of the effect exceeding the null value, quantifying directional certainty, and (2) the probability of the effect exceeding the ROPE boundaries, assessing practical meaningfulness.

To explore potential interdependencies among neuroprotective biomarkers, Bayesian correlation matrices were constructed using the correlation package (v0.8.8) with default conjugate priors. Analyses were performed separately for acute responses, defined as pre-to-post exercise changes, and for chronic responses, reflecting baseline-to-post-intervention changes. For the acute analyses, transient changes from weeks 2 and 7 were averaged to account for repeated measures prior to estimating the correlations. The Probability of Direction metric, ranging from 50% (maximal uncertainty) to 100% (complete certainty), was used to quantify the strength and direction of the associations.

## 3. Results

### 3.1. Acute and Chronic Biomarker Responses

#### 3.1.1. Brain-Derived Neurotrophic Factor (BDNF)

**Acute Responses** The three-way interaction (Session × Week × RIR) did not provide compelling evidence that proximity to failure modulated the change in acute exercise response from week 1 to week 7, yielding a modal decrease of −0.51 [95% HDI: −1.00, 0.07], with probabilities of 96.4% and 0% for exceeding null and ROPE, respectively. In contrast, the main effect of Session provided compelling evidence suggesting exercise-induced BDNF elevation, yielding a modal increase of 1.26 [95% HDI: 0.37, 1.97], with probabilities of 99.75% and 63.6% for exceeding null and ROPE, respectively. Complete model parameters are presented in Table 2, with visual representations shown in Figure 1.

**Chronic Responses** The two-way interaction (Time × RIR) did not provide compelling evidence that proximity to failure influenced chronic adaptation, yielding a modal decrease of −0.43 [95% HDI: −1.71, 0.95], with probabilities of 71.08% and 0% for exceeding null and ROPE, respectively. Model specifications and response patterns are detailed in Table 2 and Figure 1.

#### 3.1.2. Cathepsin B (CatB)

**Acute Responses** The three-way interaction (Session × Week × RIR) did not provide compelling evidence that proximity to failure modulated the change in acute exercise response from week 1 to week 7, yielding a modal decrease of −0.34 [95% HDI: −0.75, 0.03], with probabilities of 96.88% and 0% for exceeding null and ROPE, respectively. In contrast, the Session × Week interaction provided compelling evidence suggesting that acute CatB responses were potentiated following training adaptation, yielding a modal increase of 1.17 [95% HDI: 0.48, 1.92], with probabilities of 99.83% and 94.39% for exceeding null and ROPE, respectively. Complete model parameters are presented in Table 3, with corresponding visualizations shown in Figure 2.

**Chronic Responses** The two-way interaction (Time × RIR) did not provide compelling evidence that proximity to failure influenced chronic adaptation, yielding a modal decrease of −0.29 [95% HDI: −0.84, 0.25], with probabilities of 87.53% and 0% for exceeding null and ROPE, respectively. Model specifications and response patterns are detailed in Table 3 and Figure 2.

#### 3.1.3. Insulin-like Growth Factor 1 (IGF-1)

**Acute IGF-1 Responses** The three-way interaction did not provide compelling evidence that proximity to failure modulated the change in acute exercise response from week 1 to week 7, yielding a modal increase of 8.07 [95% HDI: −3.19, 19.67], with probabilities of 92.38% and 65.43% for exceeding null and ROPE, respectively. Model specifications and additional parameters are detailed in Table 4, with visual representations presented in Figure 3.

**Chronic IGF-1 Responses** The two-way interaction (Time × RIR) did not provide compelling evidence that proximity to failure influenced chronic adaptation, yielding a modal increase of 6.38 [95% HDI: −2.86, 16.08], with probabilities of 92.25% and 54.6% for exceeding null and ROPE, respectively. Chronic response patterns are illustrated in Table 4 and Figure 3.

#### 3.1.4. Interleukin 6 (IL-6)

**Acute IL-6 Responses** The three-way interaction did not provide compelling evidence that proximity to failure modulated the change in acute exercise response from week 1 to week 7, yielding a modal decrease of −0.04 [95% HDI: −0.55, 0.46], with probabilities of 54.73% and 0% for exceeding null and ROPE, respectively. The main effect of Session did, however, provide compelling evidence suggesting exercise-induced IL-6 elevation, producing a modal increase of 1.08 [95% HDI: 0.45, 1.66], with probabilities of 99.94% and 97.87% for exceeding null and ROPE, respectively. Complete model parameters are presented in Table 5, with corresponding visualizations shown in Figure 4.

**Chronic IL-6 Responses** The two-way interaction (Time × RIR) did not provide compelling evidence that proximity to failure influenced chronic adaptation, yielding a modal increase of 0.35 [95% HDI: −1.17, 1.89], with probabilities of 69.74% and 47.38% for exceeding null and ROPE, respectively. Chronic response patterns are illustrated in Table 5 and Figure 4.

### 3.2. Acute and Chronic Biomarker Correlations

Neither acute nor chronic analyses provide compelling evidence of correlations between biomarker combinations. The strongest observed correlation occurred between BDNF and CatB, yielding correlation coefficients of −0.16 [95% HDI: −0.49, 0.20] for acute responses and −0.2 [95% HDI: −0.53, 0.15] for chronic responses. Complete correlation matrices are presented in Table 6, with visual representations shown in Figure 5.

## 4. Discussion

The present investigation examined whether proximity to failure during RT modulates acute and chronic responses of neuroprotective biomarkers in resistance-trained men. Contrary to our primary hypothesis, we observed no compelling evidence for a dose–response relationship between proximity to failure and changes in serum BDNF, CatB, IGF-1, or IL-6. Rather, we observed acute increases in BDNF and IL-6 concentrations regardless of proximity to failure in both weeks 1 and 7. Furthermore, our findings suggest a difference in the acute CatB response between weeks 1 and 7. Specifically, CatB shifted from a pre- to post-exercise decrease at week 1 (−4.72% ± 31.3%) to an increase at week 7 (10.9% ± 51.8%), although the clinical meaningfulness of this shift remains unclear. IGF-1 did not significantly change post-exercise. Additionally, we observed no changes in resting concentrations of any biomarker nor meaningful inter-biomarker correlations. Overall, RT can elicit acute elevations in BDNF; however, in the current investigation, this response was not modulated by proximity to failure or other neuroprotective biomarkers.

The absence of a dose–response relationship between proximity to failure in RT sets and the acute BDNF response should be interpreted cautiously, given the limited statistical power of the present study, particularly for the 0 RIR group (n = 3). Moreover, the manipulation spanned only a narrow range of effort (0 to 6 RIR). It remains possible that a broader range, extending to 10 RIR, might reveal a dose–response effect and a minimum proximity-to-failure threshold required to elicit a transient increase in circulating BDNF.

Despite the absence of significant between-group differences, the present findings align with previous literature [16,17,18,35,36] demonstrating that free-weight and multi-joint RT with sufficient volume elicits an acute BDNF response. However, it should be noted that the aforementioned studies had participants perform at least a portion of sets to failure or use RM-based training close to failure. Thus, the observation that sets terminated relatively far from failure (at a perceived 4–6 RIR) can elicit a BDNF response warrants further investigation. If terminating RT sets at 4–6 RIR does elicit comparable acute BDNF responses to training closer to failure, this suggests that neuroprotective responses may occur with relatively low-effort RT. Importantly, training to failure is associated with greater session rating of perceived exertion [37] and elongated recovery time requirements compared to non-failure training [38]. Therefore, if training relatively far from failure can confer neuroprotection, this strategy may be more sustainable and involve lower discomfort for those seeking neuroprotection via RT.

The significant Session × Week interaction for CatB indicates a differential response in week 7 (10.9% ± 51.8%) compared to week 1 (−4.72% ± 31.3%), independent of proximity to failure. However, despite this Session × Week interaction, it is essential to note that the magnitude of the increase in CatB during week 7 was small (10.9% ± 51.8%) and may not be biologically meaningful. Previous research from Johnson et al. [35] failed to observe acute RT increasing CatB. It is possible that repeated exposure to RT may prime proteolytic pathways to respond more robustly to subsequent training stimuli, potentially through upregulation of CatB expression in skeletal muscle or enhanced secretion mechanisms [23]. However, we urge extreme caution when interpreting the present acute CatB response, especially given the small sample size.

The absence of changes in resting concentrations of BDNF following RT aligns with Gourgouvelis et al. [39] and Marston et al. [40], who reported no significant change in BDNF following 8 and 12 weeks of RT, respectively, despite including multi-joint exercises. Notably, Gourgouvelis et al. [39] also failed to observe a resting change in CatB. It is possible that the well-documented benefits of RT on long-term cognitive function occur independently of resting biomarker changes. Alternatively, the 6–12 week duration used in the current study and in previous research [39,40,41] may be insufficient to confer changes in circulating neurotrophins in young healthy individuals. Previous AT data have indicated that cognitive benefits can arise from cumulative effects of repeated transient biomarker elevations, wherein each acute perturbation contributes to progressive neural remodeling despite returning to baseline between sessions [15]. Similar findings occur in bone physiology, where intermittent mechanical loading and repeated elevations in osteogenic factors stimulate remodeling rather than sustained increases in baseline concentrations [42]. However, it remains unknown whether this same pattern of consistent acute increases in neurotrophins with RT is causative in the long-term cognitive benefits.

The robust acute increase in IL-6, independent of proximity to failure, in the present study aligned with established literature [27]. Furthermore, in agreement with previous RT data [35,36,41], there was no meaningful relationship between BDNF and IL-6 despite both biomarkers increasing. Therefore, even though BDNF has been previously expressed in human T-cell, B-cell, and monocyte cultures when treated with IL-6 [26], no study, to date, has observed a meaningful correlation between acute responses in BDNF and IL-6 following acute RT.

In 2020, Johnson and colleagues [35] reported that four sets of RT to volitional failure at 80% of 1 RM in the squat, bench press, and deadlift failed to elicit an IGF-1 response, similar to the present investigation. Though a recent meta-analysis [24] concluded that RT was capable of increasing IGF-1 concentration among those who trained for ≤16 weeks, this effect was primarily observed in elderly participants (≥60 years) and women. Moreover, the training status of participants may further modulate IGF-1 responses, as acute elevations reportedly occur more consistently in untrained individuals following high-intensity protocols [43]. Further investigation is warranted to determine whether an appropriate RT configuration can elicit a circulating IGF-1 response in trained young individuals.

An important limitation worth considering is that, although proximity to failure differed across groups for the primary training protocol, all participants completed a supplementary, standardized accessory protocol at 2 RIR immediately before the post-exercise blood draw. Completion of this accessory work varied by group. On average, the 4–6 RIR group completed ~75% of the prescribed accessory volume, while the 1–3 RIR and 0 RIR groups completed ~87% and ~91% of the prescribed volume, respectively. Two issues should be considered. First, because the accessory work was performed immediately before sampling, we cannot rule out the possibility that the average RIR across groups may have been more similar at the time of measurement, which could have influenced systemic outcomes such as BDNF. Second, differences in the percentage of accessory volume completed introduce an additional confounding factor, since variation in total training volume may have contributed to the observed responses.

Various other limitations also exist. The chief limitation is the low statistical power of the present study; thus, we urge caution with interpretation, particularly regarding the CatB Session × Week interaction. To date, no study to our knowledge has observed acute RT to increase CatB, and the present interaction should not be taken as strong evidence of this result. Moreover, our findings are limited to the duration of the training study and population, as it is unknown whether the results would have been similar with a longer training protocol or in a population with a different training status, age, or sex.

## 5. Conclusions

In conclusion, RT elicited acute increases in BDNF and IL-6; however, these responses occurred independent of proximity to failure. Practically, these findings suggest that individuals may achieve exercise-induced biomarker responses while training relatively far from failure, potentially avoiding the associated neuromuscular fatigue, injury risk, and significant recovery demands of failure training. Future investigations should incorporate cognitive testing alongside biomarker assessment and examine diverse populations with varying training statuses and cognitive function. Moreover, research should employ higher training volumes to determine if CatB is elevated following a bout of RT. Additionally, future investigations may consider examining the role of other biomarkers, such as irisin, in conjunction with BDNF, as when released in response to exercise, irisin may cross the blood–brain barrier and induce BDNF release in the brain [44].

## Figures and Tables

**Figure 1 biology-14-01756-f001:**
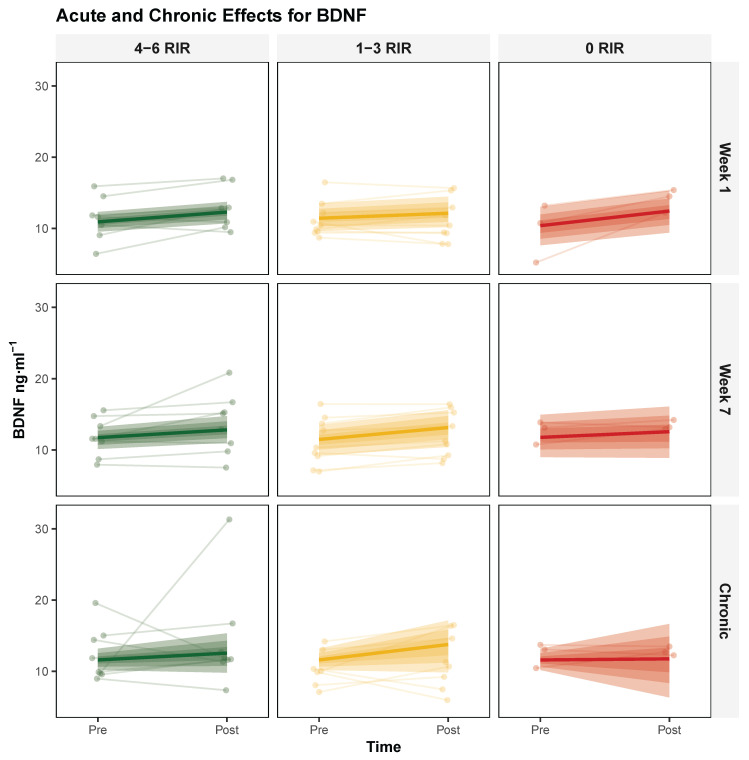
Proximity-to-Failure-Dependent Modulation Of Serum Brain-Derived Neurotrophic Factor (BDNF) Concentrations Following Acute Exercise And Chronic Training. Individual responses (thin lines) and group-level posterior modal estimates (bold lines) with 95% highest density intervals (shaded regions) for serum BDNF (ng·mL^−1^) across three exercise proximity to failure groups defined by repetitions in reserve (RIR): 4–6 RIR (green), 1–3 RIR (yellow), and 0 RIR (red). Top row: acute BDNF responses (pre- to post-exercise) at week 1; middle row: acute BDNF responses at week 7; bottom row: chronic changes in resting BDNF from baseline to post-intervention.

**Figure 2 biology-14-01756-f002:**
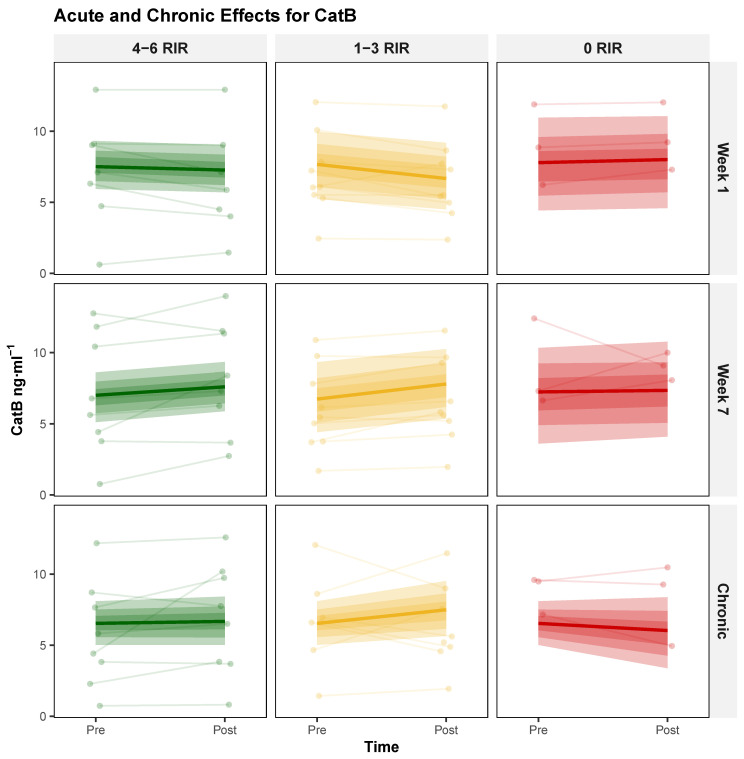
Proximity-to-Failure-Dependent modulation of serum cathepsin B (CatB) concentrations following acute exercise and chronic training. Individual responses (thin lines) and group-level posterior modal estimates (bold lines) with 95% highest density intervals (shaded regions) for serum CatB (ng·mL^−1^) across three exercise proximity to failure groups defined by repetitions in reserve (RIR): 4–6 RIR (green), 1–3 RIR (yellow), and 0 RIR (red). Top row: acute CatB responses (pre- to post-exercise) at week 1; middle row: acute CatB responses at week 7; bottom row: chronic changes in resting CatB from baseline to post-intervention.

**Figure 3 biology-14-01756-f003:**
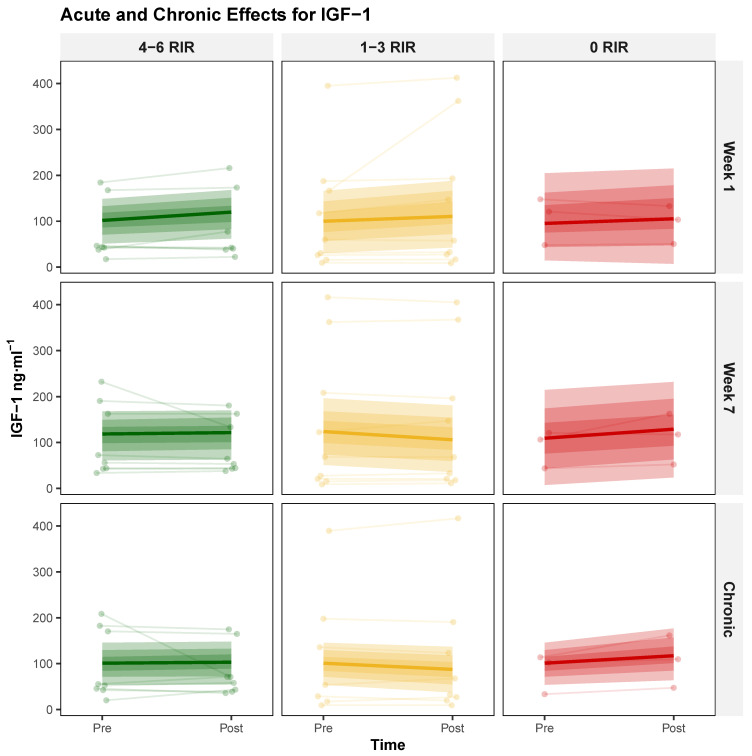
Proximity-to-Failure-Dependent modulation of serum insulin-like growth factor-1 (IGF-1) concentrations following acute exercise and chronic training. Individual responses (thin lines) and group-level posterior modal estimates (bold lines) with 95% highest density intervals (shaded regions) for serum IGF-1 (ng·mL^−1^) across three exercise proximity to failure groups defined by repetitions in reserve (RIR): 4–6 RIR (green), 1–3 RIR (yellow), and 0 RIR (red). Top row: acute IGF-1 responses (pre- to post-exercise) at week 1; middle row: acute IGF-1 responses at week 7; bottom row: chronic changes in resting IGF-1 from baseline to post-intervention.

**Figure 4 biology-14-01756-f004:**
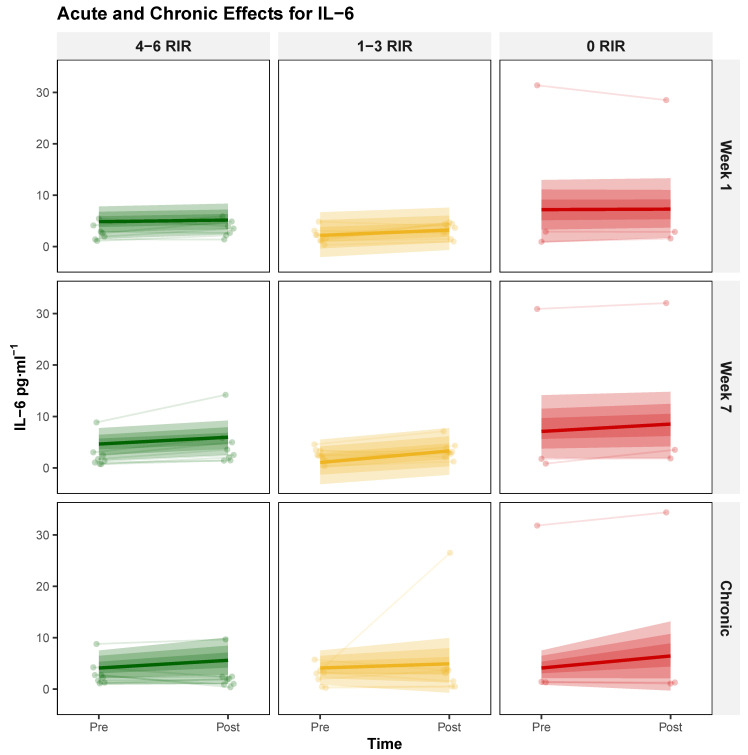
Proximity-to-Failure-Dependent modulation of serum interleukin-6 (IL-6) concentrations following acute exercise and chronic training. Individual responses (thin lines) and group-level posterior modal estimates (bold lines) with 95% highest density intervals (shaded regions) for serum IL-6 (pg·ml^−1^) across three exercise proximity to failure groups defined by repetitions in reserve (RIR): 4–6 RIR (green), 1–3 RIR (yellow), and 0 RIR (red). Top row: acute IL-6 responses (pre- to post-exercise) at week 1; middle row: acute IL-6 responses at week 7; bottom row: chronic changes in resting IL-6 from baseline to post-intervention.

**Figure 5 biology-14-01756-f005:**
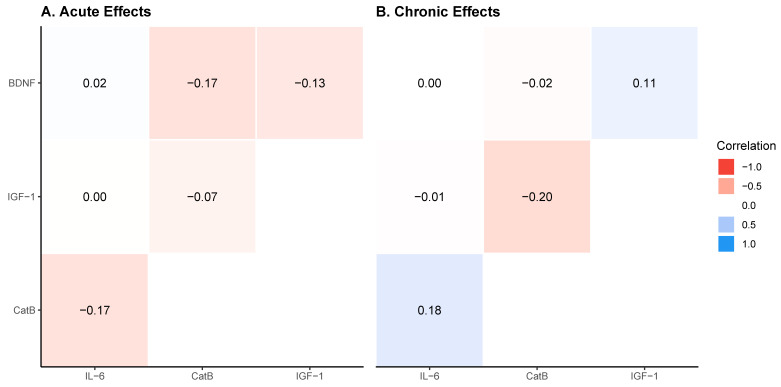
Correlation matrices for acute and chronic exercise-induced changes in circulating biomarkers. (**A**) Acute effects: correlation coefficients between pre- to post-exercise changes in BDNF, IGF-1, CatB, and IL-6. (**B**) Chronic effects: correlation coefficients between baseline to post-intervention changes in resting biomarker concentrations. Color scale represents Pearson correlation coefficients ranging from −1.0 (red) to 1.0 (blue), with values displayed in each cell.

**Table 1 biology-14-01756-t001:** Participant Descriptive Data. Data are presented as mean ± standard deviation.

Characteristic	4–6 RIR (n = 8)	1–3 RIR (n = 9)	0–3 RIR (n = 8)	0 RIR (n = 3)
Age (years)	22.50 ± 3.21	23.30 ± 3.09	21.33 ± 2.78	21.67 ± 2.08
Height (cm)	177.51 ± 5.75	174.95 ± 6.14	173.19 ± 6.23	168.03 ± 3.57
Pre Body Mass (kg)	81.33 ± 12.04	82.05 ± 12.82	78.24 ± 7.46	78.68 ± 14.37
Post Body Mass (kg)	82.61 ± 12.19	83.15 ± 13.14	79.33 ± 9.05	77.98 ± 15.49
Δ Body Mass (kg)	1.28 ± 1.64	1.10 ± 0.69	1.09 ± 2.87	−0.70 ± 2.08
Pre Sum of Skinfolds (mm)	32.20 ± 8.98	31.40 ± 11.16	29.94 ± 9.52	31.50 ± 12.49
Post Sum of Skinfolds (mm)	35.32 ± 7.18	33.70 ± 13.51	33.44 ± 9.77	35.50 ± 14.00
Δ Sum of Skinfolds (mm)	3.12 ± 3.86	2.30 ± 3.70	3.50 ± 2.59	4.00 ± 5.29
Pre Estimated Body Fat (%)	15.42 ± 3.41	15.24 ± 4.60	14.35 ± 3.18	14.90 ± 5.34
Post Estimated Body Fat (%)	16.42 ± 3.02	15.94 ± 5.31	15.46 ± 3.36	16.15 ± 5.94
Δ Estimated Body Fat (%)	1.00 ± 1.25	0.70 ± 1.16	1.11 ± 0.82	1.25 ± 1.68
Accessory-Exercise Adherence (%)	75.2 ± 19.4	86.9 ± 15.3	100 ± 0.4	91.1 ± 10.0

Abbreviations: Δ = delta; cm = centimeters; kg = kilograms; mm = millimeters; RIR = Repetitions in Reserve.

**Table 2 biology-14-01756-t002:** Posterior Effect Estimates for Acute and Chronic BDNF Responses.

Marker	Model	Estimate	Mode	Lower_HDI	Upper_HDI	P_Null	P_ROPE
BDNF	Acute	Session	1.26	0.37	1.97	99.75	63.60
BDNF	Acute	Week	0.69	−0.42	1.73	87.05	17.24
BDNF	Acute	Session × Week	0.09	−0.95	1.02	54.15	2.21
BDNF	Acute	Session × RIR	−0.02	−0.45	0.45	52.78	0.00
BDNF	Acute	Week × RIR	0.12	−0.47	0.70	64.42	0.12
BDNF	Acute	Session × Week × RIR	−0.51	−1.00	0.07	96.40	0.00
BDNF	Chronic	Time	1.18	−1.51	3.87	81.66	53.56
BDNF	Chronic	Time × RIR	−0.43	−1.71	0.95	71.08	0.00

Abbreviations: BDNF = Brain-Derived Neurotrophic Factor; RIR = Repetitions in Reserve; HDI = High Density Interval; P = Probability; ROPE = Region of Practical Equivalence.

**Table 3 biology-14-01756-t003:** Posterior Effect Estimates for Acute and Chronic CatB Responses.

Marker	Model	Estimate	Mode	Lower_HDI	Upper_HDI	P_Null	P_ROPE
CatB	Acute	Session	0.12	−0.31	0.61	74.67	2.53
CatB	Acute	Week	−0.17	−0.68	0.32	73.58	0.00
CatB	Acute	Session × Week	1.17	0.48	1.92	99.83	94.39
CatB	Acute	Session × RIR	0.03	−0.22	0.28	58.92	0.00
CatB	Acute	Week × RIR	−0.12	−0.38	0.18	77.41	0.00
CatB	Acute	Session × Week × RIR	−0.34	−0.75	0.03	96.88	0.00
CatB	Chronic	Time	0.32	−0.72	1.38	72.91	27.65
CatB	Chronic	Time × RIR	−0.29	−0.84	0.25	87.53	0.00

Abbreviations: CatB = Cathepsin-B.

**Table 4 biology-14-01756-t004:** Posterior Effect Estimates for Acute and Chronic IGF-1 Responses.

Marker	Model	Estimate	Mode	Lower_HDI	Upper_HDI	P_Null	P_ROPE
IGF-1	Acute	Session	7.02	−6.25	20.38	85.27	55.25
IGF-1	Acute	Week	7.27	−4.55	20.11	89.42	60.39
IGF-1	Acute	Session × Week	17.10	−39.55	3.24	94.29	0.00
IGF-1	Acute	Session × RIR	2.29	−5.07	9.74	74.89	16.23
IGF-1	Acute	Week × RIR	0.46	−6.35	7.30	56.62	5.54
IGF-1	Acute	Session × Week × RIR	8.07	−3.19	19.67	92.38	65.43
IGF-1	Chronic	Time	−1.59	−20.18	16.67	57.24	0.00
IGF-1	Chronic	Time × RIR	6.38	−2.86	16.08	92.25	54.60

Abbreviations: IGF-1 = Insulin-like Growth Factor 1.

**Table 5 biology-14-01756-t005:** Posterior Effect Estimates for Acute and Chronic IL-6 Responses.

Marker	Model	Estimate	Mode	Lower_HDI	Upper_HDI	P_Null	P_ROPE
IL-6	Acute	Session	1.08	0.45	1.66	99.94	97.87
IL-6	Acute	Week	−0.05	−0.77	0.75	51.93	0.00
IL-6	Acute	Session × Week	0.73	−0.23	1.63	93.55	71.88
IL-6	Acute	Session × RIR	−0.18	−0.52	0.14	87.59	0.00
IL-6	Acute	Week × RIR	0.26	−0.17	0.67	88.30	18.93
IL-6	Acute	Session × Week × RIR	−0.04	−0.55	0.46	54.73	0.00
IL-6	Chronic	Time	1.06	−1.88	3.73	77.25	66.27
IL-6	Chronic	Time × RIR	0.35	−1.17	1.89	69.74	47.38

Abbreviations: IL-6 = Interleukin 6.

**Table 6 biology-14-01756-t006:** Acute and Chronic Correlations Between Neuroprotective Biomarkers.

Model	x Variable	y Variable	r-Value	Lower_HDI	Upper_HDI
Acute	BDNF	IGF-1	−0.14	−0.47	0.22
Acute	BDNF	CatB	−0.16	−0.49	0.20
Acute	BDNF	IL-6	0.01	−0.35	0.36
Acute	IGF-1	CatB	−0.08	−0.42	0.29
Acute	IGF-1	IL-6	−0.00	−0.35	0.35
Acute	CatB	IL-6	−0.16	−0.50	0.19
Chronic	BDNF	IGF-1	0.11	−0.25	0.45
Chronic	BDNF	CatB	−0.01	−0.38	0.36
Chronic	BDNF	IL-6	0.01	−0.38	0.37
Chronic	IGF-1	CatB	−0.20	−0.53	0.15
Chronic	IGF-1	IL-6	−0.01	−0.37	0.36
Chronic	CatB	IL-6	0.19	−0.18	0.53

## Data Availability

The complete dataset, analytical workflows (including raw data, model specifications, posterior distributions, and visualization scripts), and detailed training protocols (including session-by-session prescriptions and load progression schemes) supporting the findings of this study are publicly available via the Open Science Framework at (https://osf.io/hmjrg/?view_only=6d06b111bc4e446eab41fad7c862c4ed (accessed on 5 October 2025).

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
