# Peer review of "Effect of Proximity to Failure in Resistance Training on Circulating Levels of Neuroprotective Biomarkers"

_biology, 2025, doi:10.3390/biology14121756_

Round 1

Reviewer 1 Report

Comments and Suggestions for Authors

In the current study, Benitez and colleagues demonstrate that resistance training (RT) in 38 resistance-trained men undergoing 4 different levels of efforts elicits acute increases in systemic BDNF and IL-6 levels. Moreover, the Authors report that these changes occur independently from proximity to failure. They concluded that “individuals may achieve exercise-induced biomarker responses while training relatively far from failure, potentially avoiding the associated neuromuscular fatigue, injury risk, and significant recovery demands of failure training.”

The study addresses important questions and has been nicely executed; however, there are some limitations (in addition to the small number of recruited athletes) that need further attention.

Specific Concerns

  • It does not provide a good coverage on the last achievements concerning the role of BDNF/TrkB in governing heart function. For instance, recent studies (in mice) have shown that chronic exercise increases BDNF content in myocytes (Yang X et al., Cardiovasc. Res. 2023). BDNF is needed for proper myocardial contraction and relaxation (Feng N. et al., PNAS, 2015) and to protect it from ischemia (Cannavo A. et al., Circ. Res. 2023). Therefore, assessing the systemic levels of BDNF does not necessarily anticipate (or predict) what could occur at the heart muscle level. This eventuality should be considered/discussed. Along with the fact that – when considering cognitive capacities – the lack or deficiency of BDNF/TrkB characterizes the heart as well as the brain of subjects with obesity abd suffering from psychosocial stress (Agrimi J. et al., Ebiomedicine, 2019).
  • The choice of enrolling only male resistance-trained individuals is not justified. I encourage the Authors to include a rationale backing up their choice.
  • Did the Authors look after possible changes in irisin? It may have BDND-dependent and independent effects but there is no doubt that irisin is elevated after training, and even resistance training; see, for instance, doi: 10.1016/j.exger.2015.07.006. Epub 2015 Jul 13.
  • The Discussion does not place the findings with cathepsin B in any physiologically relevant scenario.
  • Finally, the iconographic apparatus could be enlivened by the inclusion of a graphical abstract, stressing in particular the “proximity to failure” aspects of the study.

Reviewer 2 Report

Comments and Suggestions for Authors

Dear Authors

I appreciate the thorough research and the clear organization of your ideas. As a reviewer, my goal is to provide constructive feedback to help strengthen your article and enhance its impact. While reading your article, I noticed a few areas that could benefit from further development. You will see a list of comments following. I encourage you to consider these suggestions as you continue refining your work.

-----------------------

Section/subsection: Participants and Sample Size Justification

Comment/Question: In case of inclusion and exclusion criteria:

  • One of the inclusion criteria must be participant filling consent form.
  • Indeed, exclusion criteria are those by which a participant must not continue participation. In this study six participants had exclusion criteria and discontinued their participation.

-----------------------

Section/subsection: Table 1

Comment/Question: In Table 1, there are parameters in which their SD is much greater than of their mean. For example, delta body mass in group 0-3 RIR. this means a huge diversity among participants in response to exercise. It arises question whether the protocol has been fulfilled completely or supplementations was not correct.

-----------------------

Section/subsection: Blood Sampling and Analysis

Manuscript: Blood samples were collected immediately prior to pre- and post-testing sessions (weeks 1 and 8) and immediately before and after the first training session of weeks 2 and 7.

Comment/Question: Two question about blood collection:

  • What was the posture of participants while blood was collected, seated or supine?
  • How many hours was the blood collected before and after the first training session of weeks 2 and 7?

-----------------------

Section/subsection: Statistical Analysis

Comment/Question: In case of statistical analysis, the first statistical test before conducting any other test is to determine if data distribution is normal. When this was done, other tests (parametric or nonparametric) are used accordingly.

-----------------------

Reviewer 3 Report

Comments and Suggestions for Authors

Aging is associated with a decline in memory and information processing speed. At the same time, aerobic and strength training can mitigate age-related cognitive decline. The authors examined acute and chronic changes in brain-derived neurotrophic factor, cathepsin B, insulin-like growth factor-1, and interleukin-6 in four resistance training protocols varying in their proximity to failure. They showed that exercise-induced increases in neuroprotective biomarker responses can be achieved with training relatively far from failure, potentially avoiding the neuromuscular fatigue, injury risk, and recovery requirements associated with training to failure. The authors have done significant work, and the manuscript they have submitted undoubtedly has scientific and practical value. The strength of this manuscript is the logical construction of the authors' experiment and the hypothesis they proposed. The manuscript is written in accessible scientific language. There are minor recommendations that the authors should consider:

  1. Line 15. You don't describe all the important molecules studied, but then write about cathepsin. You should probably indicate which molecules in parentheses (line 24), or remove cathepsin B (line 28), or delete the sentence about cathepsin.
  2. I recommend numbering each section and subsection of the manuscript. This will improve the manuscript.
  3. According to the journal's rules, abbreviations must be expanded upon their first use in each of the three sections: abstract, main text, first figure, or table. You are using excessive abbreviations with their expansions in your manuscript. I recommend avoiding repeated use of abbreviations and their expansions. Perhaps the abbreviations should be removed here - lines 343, 367, 392, etc. Please edit text of your manuscript to comply with the journal's rules.
  4. I recommend combining the subchapters (lines 288, 302, 322, 332) with the subchapter Statistical analysis (line 279).
